# E-Mobility and Batteries—A Business Case for Flexibility in the Arctic Region

Bernt Bremdal [1,2], Iliana Ilieva [1,*], Kristoffer Tangrand [2] and Shayan Dadman [2]

1 Smart Innovation Norway, 1783 Halden, Norway
2 Department of Computer Science, UiT The Arctic University of Norway, 8514 Narvik, Norway
* Correspondence: iliana.ilieva@smartinnovationnorway.com; Tel.: +47-415-029-61

**Abstract:** This paper provides a method for determining the economic incentives and limitations for a battery used for peak clipping, with the goal of finding an optimal mix between the battery's power density and energy density. A ratio called the R-factor has been introduced, which helps determine the energy demand to curb the peak. The paper's results embrace different investment scenarios showing what battery capacity can be expected, dependent on interest rates, payback time and potential savings in power tariffs due to curtailment. In addition, the paper introduces the "wrench and cut" concept, which can help improve the investment case for batteries by combining battery operations with standard demand response operations. In particular, the effect of using a limited form of demand response-based load deactivation together with a battery has been analyzed. The investigation provided raises a point that battery degradation must be taken into account to prevent the reduction of battery life and possibly the needed payback period. The ultimate target of the presented research refers to vehicle-to-grid/vehicle-to-building developments in the Arctic region, where a vehicle is considered a mobile battery and where flexibility can be delivered in a cost-efficient way.

**Keywords:** batteries; flexibility; investment; demand response

## 1. Introduction

The transition to renewable energy is a global target well specified in the sustainable development goals [1]. Yet, in areas with severe climate, it may be particular challenging to facilitate the green shift [2]. During the last years, flexibility has been increasingly considered as a tool that can solve various grid challenges and contribute to grid resilience [3]. In particular, the use of e-mobility and batteries to provide flexibility at a local level has been an increasingly discussed option for flexibility improvements [4,5]. Furthermore, a case from the Nordics, [6], has shown that local flexibility sources (such as stationary batteries) can alleviate disturbances caused by the charging of vehicles, while in [7], the possibility for a flexibility-enhancing charging station to mitigate grid challenges has been discussed.

To contribute further to the research on green transition (also in a Nordic perspective), and in an effort to solicit fossil-fuel alternatives, this paper focuses on flexibility by discussing the use of batteries and considering the specificities related to renewables, e-mobility and battery charging. Of particular interest has been to reflect on what impact and support electric vehicles (and also electric snowmobiles) can provide together with local, renewable energy production. The idea has been to use electric mobility for load shaving during extensive periods of the year.

To achieve its goal, this research looks at cost aspects, value stacking and climate impact, as well as the aggregated effects of controlled fleet management of idle vehicles that are electrically powered. Two different sites have been the focus of case studies that have provided empirical support for the work presented. One site is located at Longyearbyen at Svalbard, Norway. Another can be found at Lehtojärvi, not far from Rovaniemi in Northern Finland.

The main research questions targeted in this paper explore the following: the possibility of balancing the battery investment versus the economic gains resulting from curtailments of power peaks and what considerations must be taken into account; the opportunity to utilize the results associated with shaving and batteries to accelerate vehicle-to-grid/vehicle-to-building (V2G/B) developments.

The first of the above-presented research question topics relates to the battery as an energy buffer as well as a power device. Payback stemming from peak load reductions is usually a matter of €/kW. Here, the energy density of the storage device determines the cost that must be justified by means of the power density, defining the return of investment and the payback time. A general method that addresses this is presented in the paper, supported by simulations.

Furthermore, since the majority of vehicles have battery costs as an integral part of the overall cost of the vehicle, the research presented here is more concerned with the yield that the vehicles can produce in terms of energy flexibility compared to stationary batteries. In a situation where the vehicles come with replaceable batteries, the presented base case (as related to the battery as an energy buffer as well as a power device) is immediately applicable.

To cover the research questions described, the rest of this paper is to be structured as follows: Section 1.1 proceeds with a state of the art, providing recent developments on the topic. Section 2 describe the methods applied. Here cost-benefit analysis, rainflow method to determine the battery's lifetime and investment studies are referred to. Section 3 presents the results from the carried analysis, while in Section 4, the results are discussed. Section 5 provides the conclusions.

### 1.1. State of the Art

During the past decade, the world has witnessed a rapid development in both battery technologies and battery applications. Battery energy storage systems (BESS) have already proven to be commercially viable and are considered for a number of business cases: in the high voltage system, at medium voltage and at the low voltage end, close to the consumer. A comprehensive overview of the possible uses of BESS has been provided in [8]. Batteries have found their place in realizing the smart grid, as energy buffers, back-up systems and flexibility instruments. BESS have become essential in microgrids dependent on supply based on renewable resources. Batteries also constitute the most essential component in the transformation from fossil-driven vehicles to electric.

Massive investments in battery manufacturing and steady advances in technology have generated significant momentum in how energy systems are to be organized towards 2030. A good overview of these developments can be found in [9]. Along with that, cost per kWh for batteries has come down too. According to [10], prices from 2010 and 2016 dropped more than 70%. Towards the end of 2022, prices have again increased due to the geopolitical situation and a surge in demand, thus coming to a pre-tax cost of €142/kWh [11].

Yet, the above-described trends do not imply that all potential business cases are economically sustainable. In this discourse, focus has been placed on BESS as a tool for energy flexibility to reduce capacity constraints in the electric infrastructure by curtailing absolute peak loads due to high concurrent consumption or extensive feeds from non-controllable, intermittent renewable resources such as solar or wind power. Multiple references, such as [12–14], have addressed the use of batteries for peak shaving or peak shifting, in order to regulate the loads in the grid. Most of these approaches place emphasis on the technical aspects associated with the use of batteries as flexibility instruments. Methods that target frequency regulations, voltage control and load balancing have been proposed. Similar emphasis has been placed on batteries "on wheels", typically referencing operations involving V2G. Along with the increase in electric vehicles (EVs), the V2G technology market is expanding too. According to [15], this market will grow from €1.67 billion in 2021 to €16.43 billion in 2027.

A principal role of V2G and associated technologies is that of regular battery systems, namely, to ensure energy operations within capacity or budget limits. A good overview of V2G applications can be found in [16,17]. A major issue associated with batteries, and especially with car-mounted batteries, is battery life, measured in terms of charging and discharging cycles. Battery health is a function of charging frequency [18]. Range anxiety has been a common phenomenon among owners of EVs. A similar anxiety is associated with degradation of the vehicles' batteries over time. Hence, lack of willingness to participate in V2G regimes for flexibility purposes has been considered an obstacle for widespread adoption of V2G operations [19]. Another issue that arises from this is what compensation to offer in return for such willingness. Choice experiments have been conducted to determine preferences associated with V2G applications [20]. The research presented here also addresses preferences, needs, requirements and benefits associated with stationary batteries and V2G for peak shaving purposes. It also offers new insight based on an economic utility approach that relates to the potential gain that can be achieved for an owner of a battery or an electric vehicle. For a stationary BESS, it is the potential gain, determined by reduction in power (kW), that determines the investment of the battery, measured in kWh. For a battery accommodated in a vehicle, the investment is associated with the driving function and not energy flexibility purpose. Hence, the gain and benefits must be compared to the loss of inconvenience and battery degradation, which can be considered a negative utility.

## 2. Materials and Methods

### 2.1. Specificities of the Flexibility Needed

When considering methods and tools for smoothing loads locally, one must look at the relationship between energy density and power density. For storage devices as an energy flexibility medium, charging time and discharge frequency must also be taken into account. This applies to electrochemical as well as thermal devices. The optimal ratio between power density, energy density and charging and discharging largely determines the requirements for the optimal investment for a given case. This can be determined mathematically if the load/consumption curve is known.

Figure 1 shows a typical consumption case for a passive house or a regular apartment (typically supplied by district heating to cover the thermal base load) with a small base load. Here the ratio $R = y/A$ will be high. Figure 2 shows a consumption curve that will apply to older detached houses with electric heating. Here, R is low. The significant bottom load creates round and elongated peaks. R is essential for the investment in batteries used for peak load curtailment.

Thus, the scope of this article is directed towards the utility of batteries for flexibility purposes. In particular: what should be catered for to ensure that a battery investment is capable of achieving flexibility gains? There are certain cases that are better suited to battery-supported flexibility services than others, and this can be determined by the R-factor. As will be concluded in the course of the paper, a low R-factor would suggest a poor case for tackling peak shaving with a battery, at least when economy is considered. However, as further indicated in Section 3.2, a poor case can be improved by "wrench and cut".

### 2.2. Demand Response and Batteries as a Means to Curtail Peaks and Reduce $CO_2$ Emissions

By analyzing advanced metering system (AMS) data for a given household or building, the average seasonal R-ratio can be established. The R-ratio is an expression of the relationship between a battery's required power density and the energy density of the battery. This provides an opportunity to determine the most economical storage concept and thus customize a good solution by means of a cost-benefit analysis. If a customer is power-tariffed, the goal will be to convert the storage function's capacity into power-reducing measures. But if the peak is permanent over time, exponentially, more energy is required because A is squared with time. Both absolute and relative metrics can be

established. The latter allows smart-charging and V2G models to be compared to other storage concepts such as a hot water tank (electrically heated) or battery.

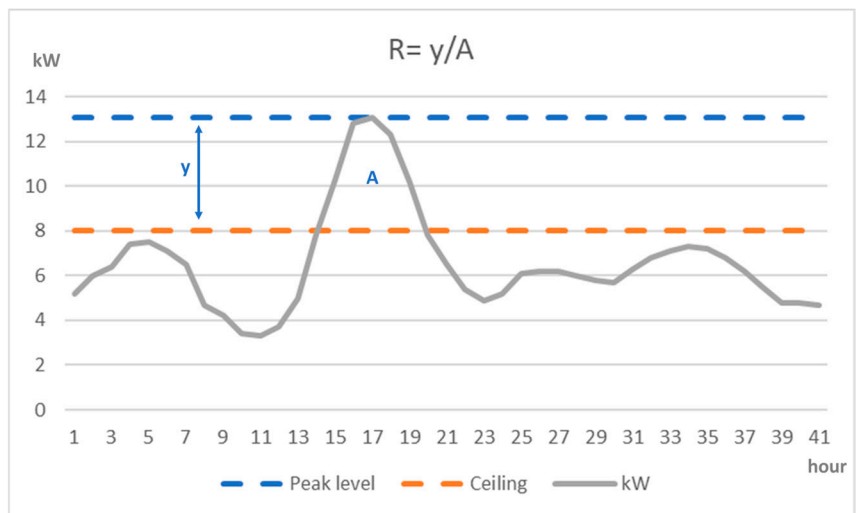

**Figure 1.** Consumption curve with high, pointed peaks that require a relatively high power density compared to energy density; R is high. (R = y/A).

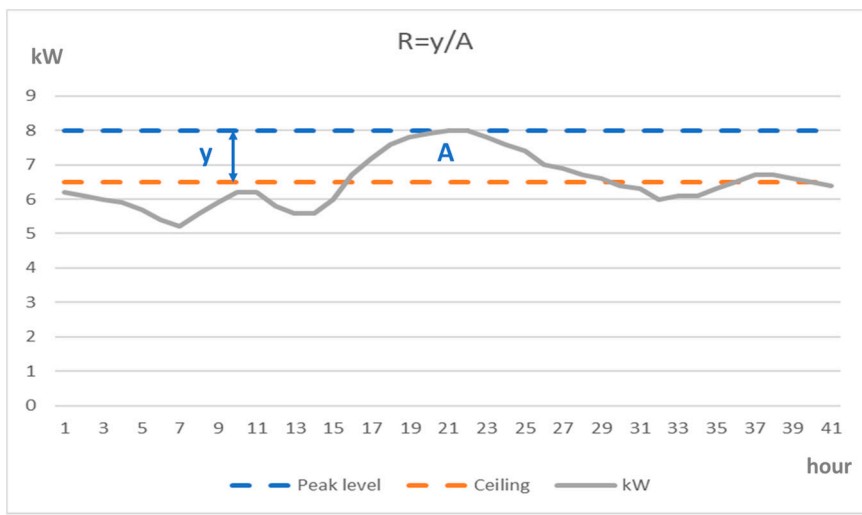

**Figure 2.** Consumption curve with low, round power peaks that require relatively high energy density in relation to power; R is thus low.

Traditionally, demand response has offered the most affordable means of flexibility management, as the investments required are quite low compared to assets such as batteries. However, demand response regimes have proven hard to implement and maintain as reliable instruments of flexibility over time since they can affect fundamental consumption patterns and the lives of people. Since demand response can inflict directly on people's lives, sufficient incentives for the participants are required [21].

Despite the cost of batteries, they offer the possibility of "service stacking" that allows the owner to use them to generate multiple revenue streams that can reduce the payback time. In addition, batteries are unlikely to impose on peoples' lives in the way demand response regimes do. Mobile batteries can be categorized in the same way. Here, any form of energy flexibility support needs to be stacked on top of a primary function or service. This affects the capitalization factor in a positive way, as long as the flexibility related service does not jeopardize the primary function—namely, to be mobile when needed.

A word of caution needs to be added here. Neither V2G/B nor smart charging can be considered a consumption-reducing measure. They are power peak shifters. On an overall level, there is no net gain or loss in terms of energy use. The energy that is discharged is essentially the energy that was used to charge it. An exception can be made for V2G/B devices if the vehicle charges the battery in locations different from the place of discharge. In that case, V2G/B can be considered an energy-saving device for one economy while a cost-imposing device for another. Yet, if the two locations are found within the boundaries of the same grid or production system, there is still no energy-reducing gain. Hence, it is important to understand that economic gains in terms of energy savings or delayed consumption in such cases are absent or very marginal. However, other economic gains for energy flexibility can be established where time-of-use (ToU) pricing is used, or with tariffs that include a cost for power, measured in euro per kWh/h. Under tariff regimes where power charges are imposed or where facilities are connected to a flexibility market, economic gains by means of load shifting can be applied. In the work presented here, such a regime has been relevant for the pilot case at Lehtojärvi in Finland, but not for Svalbard in Norway.

Yet, for Svalbard, an additional concern can be resolved by means of energy flexibility. There, energy flexibility aggregated from several sources can significantly help to reduce climate gas emissions. Since this Arctic community is still dependent on coal-fired electricity supply, the emissions caused by the electricity production are very high. Moreover, the existing power plant is getting old, and the demand for energy has increased over the years. To cover for the peaks, coal is burned at full rate year round. In addition, increasing amount of reserve power offered by six diesel generators is used when the plant's power capacity is insufficient.

However, all power plants dependent on vapor production to generate electricity are inherently slow to regulate. To produce sufficient vapor pressure to drive a turbine can take several hours if dependent on a cold start. Unlike a hydroelectric plant, where the inflow of water that drives the turbine can be regulated up and down in an instant, coal-fired power plants usually need more time. A way to compensate for this sluggishness is to maintain high vapor pressure and adjust the valves that control the turbine. By injecting more or less vapor onto the turbine blades, the electricity generation can be regulated up and down with an acceptable response time to meet the demand. Yet, this practice means that vapor must be produced so that it can satisfy peak demand at any time. Hence, it is current practice to burn coal to cater for vapor that needs to drive the turbine at maximum demand, even if that only happens occasionally. Thus, the $CO_2$ emissions are high. Power peaks handled in a different way could yield significant emission reductions. This is not only true for the considered location in the North but is relevant for many places in the world where energy supply is dependent on coal. Moreover, it offers relevant alternatives to capacity markets dependent on fossil fuel to cater for abrupt changes associated with intermittent energy production (i.e., wind generators).

If it is possible to reduce the peaks (or the maximum deviations) through demand response, batteries or other similar means, the need to drive turbines hard with sufficient vapor pressure is reduced. This means that the coal-burning rate will decrease too. Therefore, it makes sense to focus on the occasional maximum peaks and reduce those by using reserve power stored in mobile or stationary batteries. The usual means of $CO_2$ emission cuts is to reduce energy use or replace all fossil fueled plants with renewable sources. Since peaks are temporary and usually last less than a day, instruments that can curtail them are most often less costly than the means needed to slice off the persistent base load.

For the reasons described above, an investment case for peak clipping can be a good alternative. Figure 3 illustrates the concept for Svalbard. The upper chart illustrates that a few peaks of the demand curve penetrate the level of max capacity of the turbines. The peaks must be clipped or catered for. The usual practice has been to activate the diesel generators. The middle chart shows the activation of these as vertical bars. A maximum coal-burning rate is shown too. In the lower chart, the peaks have been clipped and the

valleys in the demand filled by using a flexibility strategy. The peak clipping makes the diesel generators superfluous. By clipping further, the coal-burning rate can be reduced and, thus, emissions decreased.

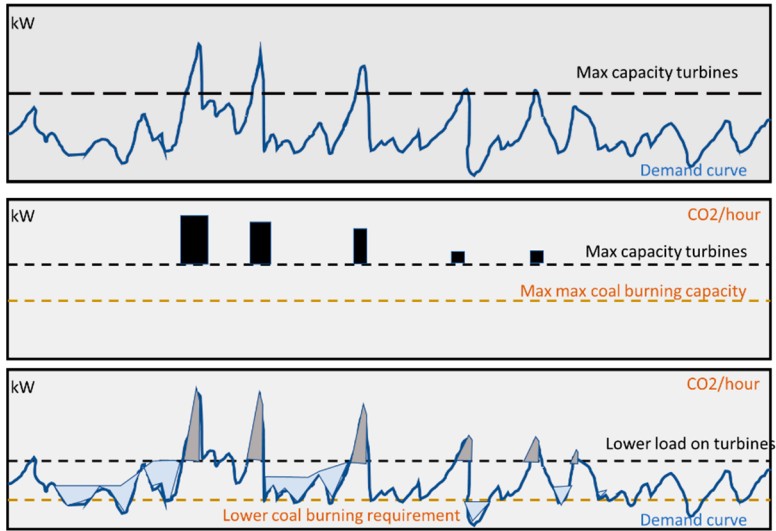

**Figure 3.** The impact of energy flexibility on $CO_2$ emissions. Upper: A load curve showing peaks that extend beyond the capacity of the existing turbines at plant. Middle: Corresponding chart showing the need for generator set activation to cater for the consumption peak. The plant is burning coal at maximum rate to support maximum electricity generation for turbines. Bottom: Energy flexibility caters for peak shaving and valley filling. Turbines do not need to cover for the highest peaks and the resulting coal-burning load can be reduced.

### 2.3. Creating an Investment Case

Storage as a flexibility device or demand response instrument has seldom carried any energy-saving potential. Hence, for the investment case that will be addressed here, only the power-reducing aspects (as associated with economic gains) should be considered. From this, it can be deducted that savings can only be achieved under a tariff regime that includes a power tariff part or a ToU tariff. Economic gains through arbitrage-oriented control of the storage are a possibility, but the price volatility in the Nordpool market has been marginal until the early months of 2021. For most storage concepts, arbitrage is not a stand-alone option but can be seen as an add-on possibility once a concept is justified by a ToU or power tariff.

A net present value (*NPV*) consideration will be used here to show how a potential investment case for batteries can be established. To start with, stationary batteries are envisioned when creating the investment case. Yet, the approach is considered valid for mobile batteries as well and, later in this work, its applicability towards V2G/B applications is the focus. The regular *NPV* equation is a sum of discounted future earnings minus the investment. If *NPV* = 0, then the future gain simply becomes a payment plan for the investment made. That is exactly what the objective is, here.

$$NPV = \sum_{k=0}^{N} \frac{Gain_k}{(1+i)^k} - Investment \qquad (1)$$

Since it is not possible to specify the monthly or annual savings with a battery without a concrete case and sufficient details about the investment, an average monthly saving will be used. The savings (or the earnings) will be based on what curtailment per month or per year the battery investment should contribute. Curtailment per period would, of course, be linked to the local grid tariff or ToU pricing applied. "Monthly" is the most common tariff period in the Northern countries and considered best suited for the targeted research.

Assuming that a loan would finance the procurement of the battery, the gain per month from curtailment would need to cover the monthly cost of that loan. The logic would, of course, be that the more you are able to cut peaks to save money, the bigger the investment that can be made. Then, the *NPV* approach can be translated into annuity-based *NPV* consideration. The expression for this would be:

$$I \leq \frac{G \times \left(1 - (1+i)^{-n}\right)}{i} \tag{2}$$

Here, *I* is the maximum investment possible and G is the monthly savings (tariff reduction, arbitrage, etc.) caused by the investment, *i* is the discount rate per month and *n* is the repayment period (in months) used.

The cost of a storage unit is usually proportional to the capacity; e.g., €500 per kWh is a typical unit cost for a battery in 2020–2021. Naturally, such cost figures could vary significantly as the storage market is quite dynamic, which, in turn, makes investment cases harder to determine.

For most common cases of peak clipping, often regardless of which purpose it needs to fulfill, it is important to sustain curtailment for an extended period. This is necessary to assure a lower bill when charged according to the use of power (kWh/h). In the case of a demand response program, a contract will require that the responsive party sustain curtailment for a predefined period (which often inflicts some negative side effects, e.g., inhibition of appliances or temperature reductions). The storage capacity needed expresses the required energy density. It becomes a function of the power ceiling imposed, the consumption curve without curtailment and the duration of curtailment. This can be expressed as a capacity:

$$Ec' = \int_{t=0}^{T} y(t)dt \tag{3}$$

Here, *y(t)* is the consumption above a load ceiling. Such a ceiling determines the maximum acceptable load, where any load about the ceiling should be cut off. Naturally, the lower the ceiling, the more likely the need for peak curtailment activation. *T* defines the period for the required curtailment. A lower ceiling will often imply the need for an extended duration of storage engagement. The period *T* can extend across a series of peaks separated by shorter relief intervals where the consumption falls below the desired threshold. In some instances, it makes sense to recover the state-of-charge (SOC) for shorter intervals than an hour. Opportunistic charging could possibly be applied to take advantage of this relief interval. However, there are some caveats associated with battery aging that need to be taken into consideration and which will be discussed later. It also depends on the recharging capabilities of the battery system and other facilities for same. Finally, it depends on the resolution of measurements defining the records on consumption. For most commercial cases, the peak recorded is the average across an hour.

Since the battery also needs to compensate for losses in the storage system, one must take these into account. Efficiency issues are often associated with the battery inverter. Hence, we have:

$$Ec = \frac{1}{c_e} Ec' \tag{4}$$

$$Ec = \frac{1}{c_e} \int_{t=0}^{T} y(t)dt \tag{5}$$

The requirement for capacity can be expressed numerically as:

$$Ec = y \times T \times cp \tag{6}$$

Here, *T* is the required peak shaving duration, *y* is the power curtailment requirement and cp is a coefficient that determines the acuteness (shape) of the curve. The acuteness can easily be calculated by analyzing the time series for energy demand. In its simplest form,

with a one-hour resolution, the peak will form a triangle and cp = 0.5. When a train of peaks follow each other in a time series, we obtain a ridge that looks like the example shown in Figure 4. Such a ridge would approximate part of the time series for the aggregated load to be controlled. The base line would represent the maximum peak level than can be allowed. The shaved-off ridge is characterized by an area that determines the necessary energy requirement for curtailment, while the maximum peak determines the maximum power needed to counteract the peak.

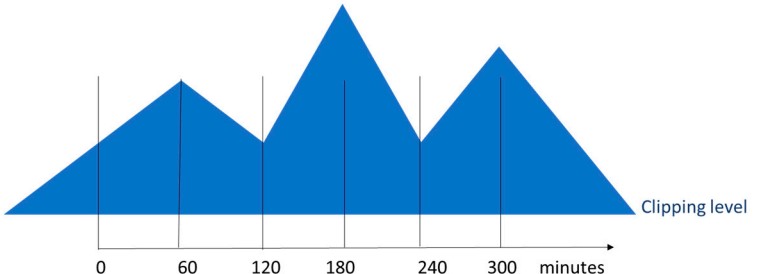

**Figure 4.** A ridge: Subsequent peaks in a time series regime with hourly measurement.

With high resolution metering, the area can be found as a function of the ridge's extension above the base line and the required period of curtailment.

$$A_{curtailment} = \sum_{t=t1}^{t=t2} y_t \qquad (7)$$

This area determines the minimum theoretical energy density needed to make the curtailment across the required curtailment period. $A_{curtailment}$ represents the peak reduction measured in kW and multiplied by the duration in hours. It determines the required energy density for the battery needed to sustain the required load reduction during $T$. This correlates directly with the cost of the battery.

$$\text{Ec} = A_{curtailment} \qquad (8)$$

The investment is thus equal to Ec × P. Here, P is the unit price for the BESS (€/kWh). This gives us:

$$I = \text{P} \times \text{Ec} \qquad (9)$$

which defines the expected cost for the storage capacity required to cater for the curtailment case represented by $A_{curtailment}$. Given that the peak clipping ambition $-\Delta y$ remains constant, the largest $A_{curtailment}$ across the full invoice period will determine the battery size. Hence,

$$Emax = \max(A_{curtailment}) \; over \; period \; T \qquad (10)$$

Since it is likely that the exact time series pattern will not repeat itself from invoice period to invoice period, *Ec* and *Emax* must be considered stochastic variables. How to calculate *Emax* will be shown later. It is important to note that the burden of defining this value lies on the grid company, or any party representing it, when defining a contract for energy flexibility. In cases where the idea is to reduce billed cost according to a power tariff, the challenge lies on the shoulders of the consuming party to save costs.

The monthly economic savings demanded by an internal budget or a flexibility contract with the distribution system operator can be defined as G.

If *Ke* is the unit cost per kW per month for the largest peak, G can be calculated:

$$\text{G} = \text{y} \times Ke \qquad (11)$$

$$P \times Emax \ \leq \ \frac{G \times \left(1 - (1 + i)^{-n}\right)}{i} \tag{12}$$

Which in turn gives the following:

$$Emax \ \leq \ \frac{G \times \left(1 - (1 + i)^{-n}\right)}{i \times P} \tag{13}$$

This determines the possible investment cases considered.

### 2.4. Determining the Battery's Lifetime

Battery life determines the maximum payback time, N, for a specific investment. If battery life is denoted l, we have the relationship:

$$N <= l \tag{14}$$

A popular method for determining the lifetime of batteries is based on the rainflow method. It is a cycle counting algorithm that was originally developed to determine material fatigue. In [22], it has been shown that battery lifetime can be determined based on charge-discharge patterns. The method has also been advocated by [23]. The battery capacity is divided into $J$, equally sized segments having an energy capacity of $e_j$. Generally, both the accuracy and the computational complexity increase with increasing $J$. One of the basic concepts of this method is the cycle depth, which is the difference between the depth of discharge at the beginning and the end of the discharge or charge half cycle. We define it both for each segment ($\Delta \delta j$) and for the battery as a whole ($\Delta \delta$).

Cycle depth at segment level:

$$\Delta \delta_{t,j} = \frac{e_{t,j} - e_{t-1,j}}{Emax} = \frac{\Delta e_{t,j}}{Emax} \tag{15}$$

where *Emax* is the energy capacity of the battery. Using segment and battery capacities, we define the cycle depth of one segment as follows:

$$\Delta \delta_t = \frac{e_j}{Emax} = \frac{1}{J} \tag{16}$$

This constant value applies for all segments. The cycle depth at battery level is obtained by summing up all the segments:

$$\Delta \delta_t = \frac{\sum_j \Delta e_{t,j}^{dis}}{Emax} \tag{17}$$

In practice, consecutive charge-discharge cycles of various sizes can be treated so that local extrema within a sequence such as $e_2$ and $e_3$ in Figure 5 are ignored and the $e_1$ and $e_4$ define the $\Delta \delta_{1,4}$ Once this is established for this sequence, the process is repeated from $t = 4$ (for the example in Figure 5).

Life-loss of a battery is thus proportional to the sum of all half cycles, discharge and charge:

$$Lifeloss = m \times \Delta \delta_t \tag{18}$$

where m is a coefficient. If a battery capacity has a nominal life of C cycles, than the remaining life can be found for the battery if it is monitored well.

By means of simulations, time series that would determine the life-loss in months, and not in cycles, can be produced. Furthermore, the life of a battery in terms of months can be estimated and used to determine if l is greater or less than $n$. What is important to note

here is that the payback on the investment, I, is dependent on how the battery is managed. If $\Delta\delta_t$ is kept small, than the payback can be extended, yielding a higher NPV.

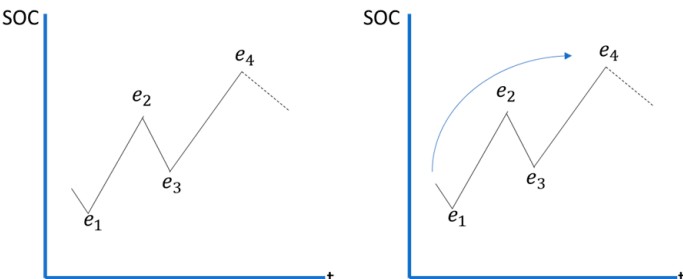

**Figure 5.** The cycle counting principle using the rainflow algorithm.

It should be noted from this that the R-factor is susceptible to the discharge/charge routines applied. The likelihood of individual peaks being characterized by a better R-factor than a continuous train of peaks is high. That caters for a smaller battery with lower investments but also a higher frequency of charge/discharge. A smaller and more affordable battery might thus be subjected to a higher frequency of deep cycles and be exposed to an accelerated life-loss that would demand a shorter payback period for the investment. A bigger battery can sustain a duration of multiple peaks in a row without recharging, until the state of charge has reached the minimum level. Hence, the frequency of deep discharge/charge cycles will be lower, which could extend the battery life and thus the duration of the payback period.

## 3. Results

### 3.1. A Business Case for Flexibility

Some individual studies have been conducted to demonstrate the method described here. The idea of investigating a business case where batteries (also from mobile applications) are utilized has been inspired by [24]. Figure 6 shows investment constraint curves based on the mathematical approach leading to the inequality 13. An investment constraint curve defines the boundary between a profitable and a non-profitable investment. It defines the relationship between the maximum battery size that can be justified based on the savings that peak curtailment can achieve and the required payback time.

To determine whether an investment in a battery is profitable for peak shaving purposes, the inequality 13 has been applied. This inequality is based on a standard *NPV* calculation, as shown in Equation (1). In addition, a perspective on the battery life is needed, as explained in the previous paragraph. If the battery life is determined to be shorter than the required payback time specified by a loan giver (or by an inhouse economic constraint), the battery life must be used as the required payback time. For the purpose of the case presented here, we have assumed a common capacity tariff where the energy user is invoiced for the highest peak throughout the period. Hence, it is necessary to discount the battery investment accordingly. This determines the value of the variable n. In this discourse, the period is one month, which is not unusual. It should be noted here that a similar approach could be applied for ToU tariffs and for peak shaving based on tariffs such as subscribed power [25], but with some adjustments. Moreover, the internal interest rate (*i*) or the interest rate specified by the loan giver must be used. Since batteries, generally, are priced according to energy capacity P (price per kWh), curtailment with tariffs like the one used here yields sole benefits in terms of power reduction, measured in kW. Savings G specify the accumulated monetary gain per curtailment, which, in turn, determines the battery life according to the rainfall method. Thus, it is possible to determine the relationship between the battery size and the size of the curtailment, as given by the inequality 13. The graphs in Figure 6 illustrate this relationship, considering different values for the variables.

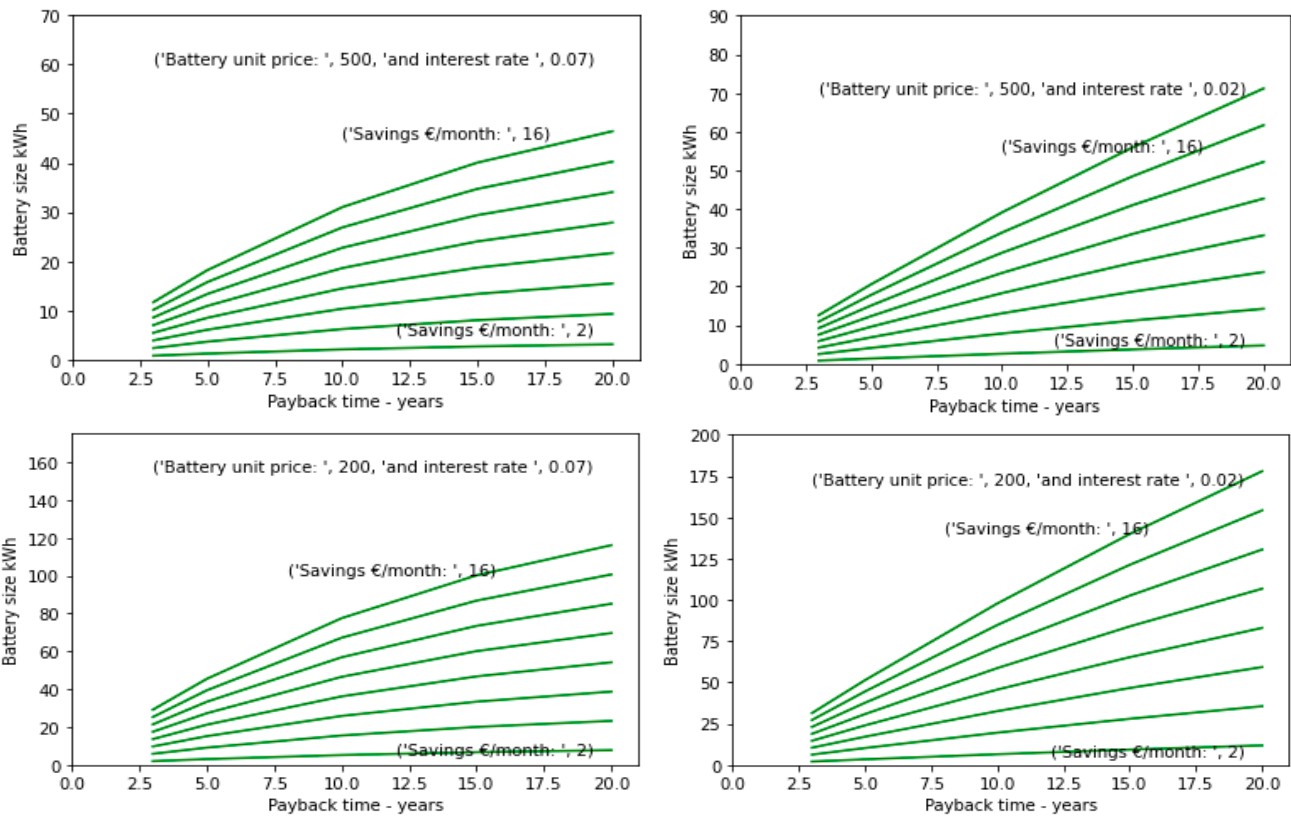

**Figure 6.** Different investment scenarios showing what battery capacity can be expected dependent on interest rates, payback time and potential savings in power tariffs due to curtailment. Upper plots: battery unit price of to €500 per kWh and interest of, respectively, 7% and 2%. Lower plots: battery unit price of to €200 per kWh and interest of, respectively, 7% and 2%.

In Figure 6, the green lines represent the attained gain (G) (i.e., the savings per month)—from €2 to €16, with a €2 interval. With potential monthly savings of €10 per month, a 10-year payback period would allow an investment in a 52.1 kWh battery if the price becomes as low as €200 per kWh with a 7% interest rate. This refers to the value at 10 payback-time years for the 5th (from bottom to top) green line (i.e., savings of €10 per month) in the lower left graph in Figure 6. Given that the unit price is closer to €500 per kWh [26], a battery bigger than 20.7 kWh will not be profitable. With a very low interest rate of 1%, the same payback period will, of course, allow a bigger battery. Considering a €200 battery price per kWh, the battery size can be increased by 25% to 65.2 kWh. This shows the impact of a lower interest rate. With the higher battery cost, the percentage increase will be more than 30%, allowing for the procurement of a battery size of 27.1 kWh. It can thus be concluded that the savings due to peak load reduction are very sensitive to the power tariff level. The more the payment for grid capacity, the better the business case. The R-factor makes a significant difference, too, as it reflects the required activation time of the battery to harvest the savings. If the duration of a peak is long, the business case becomes poorer, as a bigger battery is required to harvest the gain stemming from curtailment.

The 2% interest rate is a modest benchmark for investments but reflects recent interest rates for bank savings in several European countries. More specifically, the 2% interest rate represents an internal rate that prevailed in the 2020–2021 period. With today's internal rate of 2.5–4%, the payback requirement would be more demanding. If a higher internal interest rate requirement is maintained, the investment cases reflected in the table and the curves will obviously become less attractive. The numbers in Table 1 confirm that the peak shaving, measured in euros, needs to be well honored to defend the investment in a large battery.

**Table 1.** Investment cases for interest of 2% p.a. and 10 years payback time, as dependent on the battery's unit price.

| Average savings per month (€) | 4 | 16 | 4 | 16 | 4 | 16 |
|---|---|---|---|---|---|---|
| Unit price battery (€/kWh) | 200 | 200 | 300 | 300 | 500 | 500 |
| Max capacity of profitable battery (kWh) | 26 | 104 | 17 | 70 | 10 | 42 |

In one of the Nordic cases considered, the grid tariff included a power demand element. This has been introduced to reduce peaks and avoid too many simultaneous loads. Curtailment of the maximum peak per month with 1 kWh/h would yield a gain of €3.80 per month. According to Table 1, a curtailment measure that has a unit cost of €200 could be, at most, 26–27 kWh. That means that it would be possible to sustain clipping of a maximum peak of 1 kWh/h over a period of a full day, if needed, or manage a series of peaks in a row without charging. That could extend the required payback period. In the case of the more expensive option where the cost is €500 per kWh, a battery of only 10–11 kWh can be defended, but it would be able to sustain curtailment of 11 h if needed.

If the target savings are €16 per month for the same case, it means that curtailment would have to be in the range of 4–5 kWh/h. In such a case, the most expensive battery option would be 42–43 kWh and would allow a series of 9–11 one-hour clippings without charging. For the €200 alternative, this could be almost tripled.

The graph in Figure 7 shows how long different batteries are able to sustain different curtailment levels. If the R-factor of the $A_{curtailment}$ is low, suggesting a wide consumption curve or ridge to be eliminated by the battery, the curtailment capacity of the battery is obviously reduced. Thus, batteries become less effective as flexibility instruments as the sustainability requirement increases. The greater the battery, the more significant this issue becomes. In fact, the absolute difference in performance between a big battery and a small battery becomes asymptotically very small. As a consequence, it would be wise to combine investment in a battery with other flexibility measures. This would lead to a strategy where it would make sense to do the opposite of what regular demand side management plans often adopt, namely, to do peak shaving and valley filling.

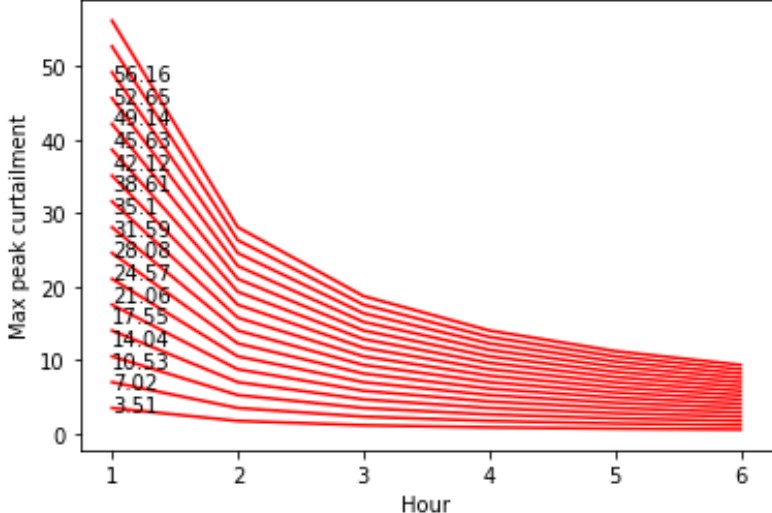

**Figure 7.** Level of peak curtailment the battery sizes referenced in Figure 6 can sustain over extended periods.

Following the reasoning above, the R-ratio for different type of crests and peaks in consumption patterns have been calculated. The R-ratio determines the required combination of power density and energy density in a battery to manage a complete curtailment of increasing duration (Figure 8).

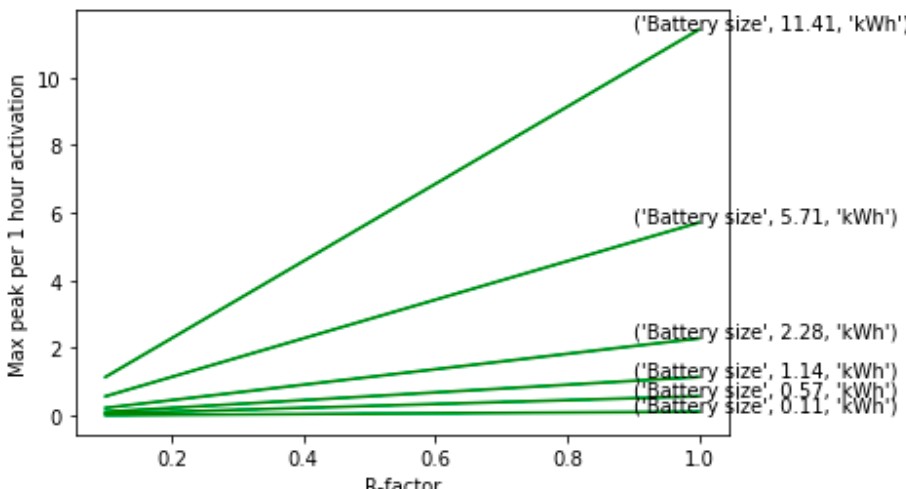

**Figure 8.** Relation between an increasing R, the battery size and the activated flexibility.

### 3.2. "Wrench and Cut"

The "wrench and cut" principle implies that demand response and battery operations should be combined if possible. This is illustrated in Figure 9. There, it is implied that instead of shaving off the load indicated by the blue line (continuous peaks and valleys, reflecting energy demand A1), it would be better to concentrate the load to a specific part and use the battery for a shorter period, in order to shave off the peak and reduce the peak load further (red dashed lines).

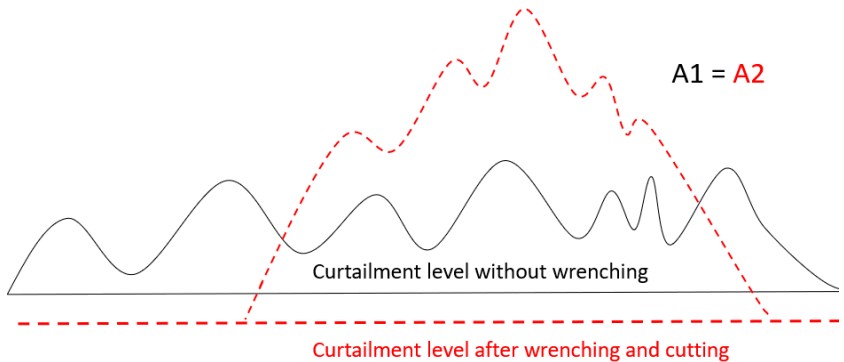

**Figure 9.** Illustration of the "wrench and cut" principle.

Using the battery to ramp up extra during a shorter period would increase the return on investment with certain power tariffs and also increase the capitalization factor for the battery. In principle, the same amount of energy is offered by the battery but for a shorter period, where the flexibility needed prior and after the discharging is covered by demand response operations. Possibly, the divided (or shorter) demand response operation can go unnoticed, as in many cases the disconnection period does not need to be very long to produce the desired effect. For instance, instead of a two-hour disconnection, two half-an-hour breaks can be introduced, with a one-hour battery discharge period in-between.

Again, it depends on the extent of the ridge and the R = y/A ratio of the original time series. Figure 10 illustrates the concept. There, the following cases have been considered:

- Case 1: Demand response is used to shave off the grey area (A1) with no battery. The gain is $\Delta y1$. This case depicts a situation where the demand response period can be so extensive that it erodes consumers' confidence and willingness to yield. In addition, it is possible that rebound effects may occur, creating new peaks.

- Case 2: The battery is used to shave off the grey area (A1) with no demand response. The gain is $\Delta y1$. The R = y/A ratio is unfavorable. It requires a larger battery, which yields little gain in return.
- Case 3: A combination of Case 1 and Case 2. Represents a combination that reduces and splits up the demand response periods significantly, and endurance of consumers would have to be less extensive.
- Case 4: Demand response is used to shave off the grey area and part of the blue (A1 + A2) with no battery. The gain is $\Delta(y1+ y2)$. The case is similar to Case 1 but affects the consumers more because larger reductions are required. Thus, their tolerance may vanish quickly. However, if the load reduction requests are for a short time, they may be more acceptable.
- Case 5: Battery is used to shave off the grey area and part of the blue (A1 + A2) with no demand response. The gain is $\Delta(y1+ y2)$. The case is similar to Case 2 but requires a much bigger battery. The R-ratio will, in fact, be reduced.
- Case 6: A combination of Case 4 and Case 5. The R-ratio of the battery would become significantly better, and the burden imposed by the demand response program could be kept within tolerance level.

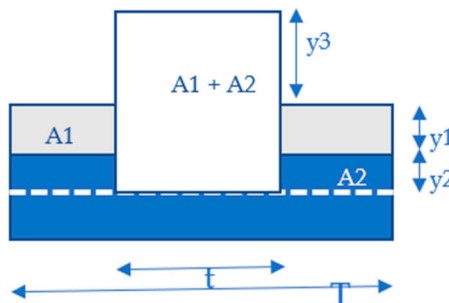

**Figure 10.** The effect of using a limited form of demand response-based load deactivation together with a battery.

Referring to the curtailment exhibited by the upper red line in Figure 9, we can find the power required by the battery. Power needed to curtail a peak and expressed in terms of kW would be given by:

$$y_c = y_1 + y_2 + y_3 \qquad (19)$$

Since it is only $y_1 + y_2$ that would yield economic benefits, $y_3$ should be kept as small as possible while, at the same time, compensate for the demand response actions that shave off loads before and after. It is important to note that for real time series, we are trying to reduce valleys and peaks that build up slowly and that, after culminating in a crest, eventually fade out. The battery's purpose is to shave off that crest, but because of a low R-ratio, it can do better with the help of demand response actions.

Using the notation from Figure 10, the following expression can be established since the required energy to cater for the intervention is constant:

$$(T - t)(y_1 + y_2) = t \times y_3$$
$$t = cT \ where \ 0 < c \le 1 \qquad (20)$$

From this, it follows that:

$$c = \frac{y_1 + y_2}{y_1 + y_2 + y_3} \qquad (21)$$

$$y_3 = \frac{y_1(1 - c) + y_2(1 - c)}{c} \qquad (22)$$

If there is no demand effort, $y_3$ would be zero and $t$ = T. If the load is wrenched and concentrated around the period $t = \frac{1}{2}T$, then $y_3 = y_1 + y_2$. In that case, the power requirement for the battery would be:

$$y_c = 2(y_1 + y_2) \tag{23}$$

The value of demand response would reduce the battery investment. In addition, there is extra power to handle the peaks $y_3$, thus providing more value. Figure 11 shows how control of both demand response and battery management systems (BMS) could be ensured. S1 is a signal that inhibits a load contributing to the aggregated consumption. S2 reactivates the load at the same time as the BMS discharges the battery. S3 suspends discharging. Most often, demand response causes load shifting. This means that power use is moved from one period to another. This tends to cause a rebound effect where the curtailment achieved in the first period is added to the general load in the second period. The effect is more concentrated energy use, which better utilizes the battery capacity. The practical consequence of this is that the intervention enabled by a direct load reduction can be made small and less intrusive for a longer period. To reestablish the normal energy situation quickly (e.g., comfort level), more power can be used for a shorter time. A significant part of this power can be provided by discharging a battery. There is seldom a net energy saving involved, but the R-factor can be increased and therefore yield a net economic utility.

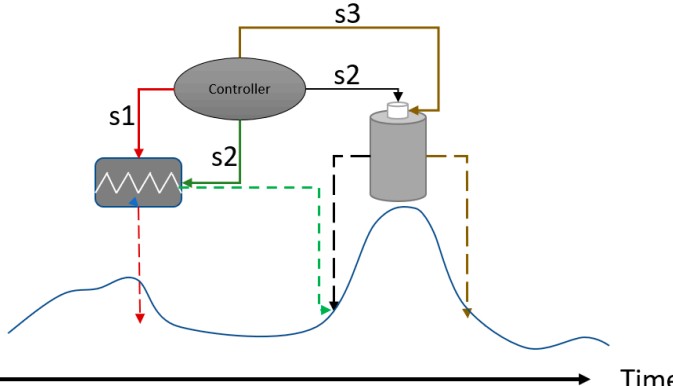

**Figure 11.** A control system that aligns demand response and battery usage. The blue line illustrates a load curve typically caused by a heater, before and after intervention. When signal s1 is activated, the load is suppressed by means of a regular demand response intervention. When s2 is activated the load suppression terminates. This causes a rebound effect; i.e., the heater works to reestablish set-point temperature. At the same time, the battery is discharged.

## 4. Relevance for V2G/B Applications

EVs can be considered mobile batteries. In addition, for the Arctic region, snowmobiles can also be utilized in the way EVs are. Specificities of two relevant types of electric snowmobiles are provided in Table 2, below. For the purpose of this article, the possibilities associated with electric snowmobiles are considered similar to what EVs with smaller battery capacity can provide. Thus, in the course of this research, battery-driven snowmobiles have fallen under the more generic "vehicles" term used.

Here, three associated research questions should be considered: Can V2G/B concepts for flexibility services be used in the same way as stationary batteries? Can they be competitive when compared to stationary batteries? What is the payback aspect? This research documented that capital investments for stationary batteries, measured in unit cost/per kWh, must be balanced against the potential gain, which is commonly based on a reduction in kW. For a replaceable battery mounted in a vehicle, similar considerations can be made. However, most batteries in vehicles are fixed and not purchased separately or dedicated to flexibility services. Moreover, they are not connected to the grid at all times like a stationary battery. The typical challenge associated with V2G/B, where vehicle

mounted batteries are used as energy flexibility instruments, are related to three principal causes—battery life-loss, inconvenience and lack of obvious business models. Life-loss of batteries treated above applies to mobile batteries too. The management issues required to handle the balance between mobility, the primary function of the EV and flexibility has been addressed in [27]. Inconvenience is typically used as an argument against the adoption of a V2G/B regime. Together with a fear of battery life-loss, concerns are raised that the freedom of EV use can be restricted, in spite of the fact that at many places EVs, as well as electric snowmobiles, are stationary for up to 90% of the time [28]. Such concerns increase with the extent and duration of the discharge of the EV battery.

**Table 2.** Types of electric snowmobiles considered relevant for the provision of flexibility.

| Snowmobile Brands | Type 1 | Type 2 |
|---|---|---|
| Battery capacity li-ion (kWh) | 23 kWh | 7–21 kWh |
| Battery peak power | 67 kW | 60 kW |
| Charging rate—onboard charger | Up to 6.6 kW AC | Up to 6.6 kW AC |
| Charging ports | Type 1-CCS | Type 1-CCS |
| Top speed | 100 km/h | 100 km/h |
| Range | 140 km | 100 km |
| Weight (kg) | 341 | 270 |
| Price | From €15,000 | From €15,000 |

The value proposition aspects will be discussed in light of what investments in stationary batteries represent. Since EVs are purchased primarily for services that usually do not include V2G/B, any other use of the vehicle can be considered as a bonus, as long as the bonus does not compromise the primary function. Value stacking is a popular ambition to increase revenue streams or personal utility by assigning more services to an artifact. The relative value of V2G/B compared to a stationary battery can be seen as the difference in investment for a given payback. The payback in terms of V2G/B needs to be defined in a contract that determines the compensation for the willingness to make the EV available for peak load curtailment.

V2G/B has been available for CHAdeMO-ready vehicles for several years. The current delivered from a battery to a building or to the grid is typically managed through a two-way charger. The typical max power is 10 kW. To illustrate what it would take to sign up an EV owner for a V2G/B regime, a use case with U.S. and Finnish eSleds has been studied. The electric sleds can accept 6.6 kW charging power. Theoretically, it would take approximately 3.5 h and a little more than an hour to charge the small and the big snowmobiles specified in Table 2. The magnitude of the reverse flow for these vehicles depends on multiple factors and is still uncertain. This has been a topic in the Smart Charge project [29] but requires further research. A discharge level between 30 and 50% of maximum charge capacity is currently what appears to be relevant.

To illustrate the concept highlighted, two scenarios have been chosen: 2 kW and 5 kW discharge capacity. This means that the biggest model listed would be able to maintain 2 kW and 5 kW discharge for approximately 7 and 2.5 h, respectively. This can again be aggregated by concurrent discharge from multiple snowmobiles under fleet management control, as described in [27]. Assuming that the power part of the local grid tariff is €3.80 per kW, the discharge of 2 kW would be worth €7.60 per activation. Hence, it is possible to calculate the kind of offers that can be made for the EV owner to take part. The starting point is the value per month to ensure that that the monthly peak is systematically clipped. If we further assume that a maximum 2-h activation of the BMS (Battery Management System) for the reverse flow would be required, a comparison with an investment in batteries can be made based on, e.g., the following assumptions: a battery

cost of €500 per kWh, 5-year payback, interest rate of 2%. The results are shown in Table 3. The nominal unit cost of a battery must not exceed €220 per kWh for a profitable investment to reach the capacity that is readily available with V2G/B. This shows that V2G/B offers a business potential similar to that of stationary batteries, since the batteries are primarily purchased for a different value-generating purpose. Both alternatives share the challenge associated with life-loss. However, as battery degradation for a stationary unit constitutes an absolute measure, life-loss for a mobile battery due to peak shaving is a relative measure. It is a percentage loss of the total number of deep charging cycles that is primarily due to the mobility function. Similar to stationary batteries, "wrench and cut" can be applied to reduce the burden on both the battery and the owner, as well as the use of the EV.

**Table 3.** Comparison of V2G/B with batteries. A profitable investment in battery is dependent on a low price to achieve the same capacity as V2G/B. Analysis is done based on different prices: €500 (case 1), €300 (case 2), €200 (case 3).

| V2G | | | Stationary Battery | | |
|---|---|---|---|---|---|
| Peak Shaving Capacity for One Hour [kW] | Monthly Value of Peak Shaving (€) | Investment Limit (€) | Price Case 1 Peak Shaving Capacity One Hour [kW] | Price Case 2 Peak Shaving Capacity One Hour [kW] | Price Case 3 Peak Shaving Capacity One Hour [kW] |
| 0.5 | 1.9 | 111.5 | 0.2 | 0.4 | 0.6 |
| 1 | 3.8 | 222.3 | 0.5 | 0.7 | 1.1 |
| 2 | 7.6 | 444.6 | 0.9 | 1.5 | 2.2 |
| 5 | 19 | 1111.5 | 2.2 | 3.7 | 5.6 |
| 10 | 38 | 2223 | 4.5 | 7.4 | 11.1 |

What is more important is the "human capital" factor. In this context, inconvenience can be termed "V2G/B fatigue". The more a membership in a V2G/B-based flexibility regime restricts the freedom of vehicle use, the more frustrations may increase and willingness to continue as a V2G/B member may decay over time. Hence, V2G/B compensation should follow the same principle as shown with stationary batteries.

Thus, the economic gain related to peak shaving each month needs to compensate for a quantity (such as life-loss), as well as for the more qualitative aspects stemming from the psychologically oriented V2G/B fatigue. Life-loss specifies an estimated reduction in years of operation due to the degradation of the battery. V2G/B fatigue is an expression for the inconvenience suffered due to the secondary use of the EV. Both can be expressed as a loss or, with reference to game theory, as a negative utility that needs to be compensated. The fatigue increases over time and can be modeled as a non-linear function. It can be compared to a decay function or a discount factor [30]. Hence, the gain from the peak clipping must outmatch the negative utility that will increase with the number of activations. This work argues that the fatigue factor can be set to be similar to the annuity in Equation (13). By using non-monetary utilities as a starting point, we can determine the initial "unit goodwill capital" of an EV owner as $P'$. This defines the willingness to reserve 1kWh of the EV's battery for flexibility purposes. The fatigue concept can be modeled according to a decay function $\rho(n)$, which decreases the goodwill over time. The decay function relates to increasing concerns over both life-loss and inconvenience. If the notion of goodwill capital is measured in utility or in monetary terms, it can be argued that the decay function can be modelled as it is for investments in hardware.

Admittedly, this is still an area of research, but it appears to be a fair approximation of the increasing concerns over capital degradation, comparable to an average value reduction of used cars in the open market or common depreciation concepts. The ordinary interest rate, $i$, can therefore also be used. This also applies because the utility can eventually be converted to a monetary value. *Emax* is the maximum energy that can be discharged from the EV battery, as addressed above. Inequality 13, which was used to determine the maximum profitable battery size, can therefore be used to design contracts for participation

in a V2G/B regime. Both the temporal extent (*n*), measured in time or number of activations per month, and the magnitude of compensation can be determined. If the Gain, as shown in Table 3, is a constraint, inequality 13 can be used to determine the "unit goodwill capital", provided that n is fixed. *P'* thus displays the marginal value loss that can be accepted. If that loss proves to be lower than the comparable value loss in the used vehicle market, the V2G/B business concept should be viable. If the indifference level between the marginal value loss *P'* due to flexibility purposes and the value loss $Loss_{EV\ value}/year$ for the vehicle as a whole is used, then, with $P' = \frac{Loss_{EV\ value}}{year}$, inequality 13 can be rewritten as:

$$\mathbf{G'} \geq \rho(n) \times E_{max} \times P' \tag{24}$$

$$\mathbf{G'} \geq \frac{E_{max} \times i \times P'}{\left(1 - ((1+i)^{-n})\right)} \tag{25}$$

This yields the required minimum compensation for taking part in the flexibility scheme. Even if *P'* is only sentiment-based, a monetary value can be established by means of a choice experiment such as [20,31]. By comparing preferences over an array of alternatives and with several human subjects, numerical values for *P'* can be determined. Such alternatives could be guarantees, depreciation rates for regular investments, the price difference between a new and a used EV, etc. The numerical result precipitating from this can be used as a basis for determining the renumeration for entering into a contract. This enables the production of the type of charts shown in Figure 6, which can make the effects of changes in the decay function and activations in the same transparent way.

## 5. Conclusions

A method for determining the economic incentives and limitations for a battery used for peak clipping (also with reference to mobile versions) has been presented. Since economic incentives stem from curtailment measured in euros per kW reduced, and unit costs for batteries are measured in euros per kWh, an optimal mix between the battery's power density and energy density must be found. A ratio called the R-factor has been introduced. This is a ratio between the height of the peak measured in kW and the area that the peak encircles, which is a function of the duration of the peak and its height, which then determines energy demand to curb the peak. It is shown that the investment case for batteries is sensitive to the R-factor. A high R-factor can improve the investment case significantly. By introducing a concept that has been called "wrench and cut", the investment case for batteries can be improved. By combining battery operations with standard demand response operations, the R-factor can be improved. Since the deactivation of loads will be shorter than without the battery, it should not impose too much on people's daily life. At the same time, battery costs can be reduced.

A point has been raised that battery degradation must be taken into account to prevent reduction of battery life and, possibly the needed payback period. Here, the rainfall method has been included in the approach presented. The rainfall method helps to determine the balance between charging and discharging to sustain the maximum life of the battery. This also decides the R-factor. All this can be integrated with V2G/B developments where a vehicle is considered a mobile battery. By using game theory, the inconveniences and the benefits addressed can be translated into utilities to determine the "indifference level". This implies the boundary between willingness to take part in a V2G/B regime or not and thus helps to determine the level of compensation needed to overcome that indifference.

The research provided in this paper opens avenues for enhanced utilization of mobile or stationary batteries, something that will have a particularly positive effect on regions such as the Arctic, where coal is still a major energy source.

**Author Contributions:** Conceptualization, B.B.; Data curation, B.B., K.T. and S.D.; Formal analysis, B.B., K.T. and S.D.; Investigation, B.B., K.T., S.D. and I.I.; Methodology, B.B.; Software, K.T. and S.D.; Supervision, B.B.; Writing—original draft, B.B.; Writing—review & editing, I.I. All authors have read and agreed to the published version of the manuscript.

**Funding:** This work has been supported by the CINELDI project (Centre for intelligent electricity distribution), an 8-year Research Centre under the FME-scheme (Centre for Environment-friendly Energy Research, 257626/E20). The presented research has been part of a CINELDI task for 2022, where business models, energy flexibility and grid impact are of main focus. The authors gratefully acknowledge the financial support from the Research Council of Norway and the CINELDI partners.

**Data Availability Statement:** Not applicable.

**Acknowledgments:** The authors are grateful to have been able to utilize results generated from the Interreg North funded project Smart Charge, which has been carried out as a cooperation between the Computer Science and Computational Engineering Department of UiT The Arctic University of Norway and The Lapland University of Applied Sciences in Rovaniemi in Finland from 2019–2021.

**Conflicts of Interest:** The authors declare no conflict of interest. The funders had no role in the design of the study; in the collection, analyses, or interpretation of data; in the writing of the manuscript; or in the decision to publish the results.

## Abbreviations and Nomenclature

| | |
|---|---|
| $c_e$ | Energy efficiency factor |
| $A_{curtailment}$ | Peak reduction measured in kW multiplied with the duration in hours |
| $e_{t,j}$ | Energy capacity at time $t$ for segment $j$ |
| $\Delta\delta$ | Cycle depth for the battery as a whole |
| $\Delta\delta j$ | Cycle depth (the difference of the depth of discharge at the beginning and the end of the discharge) for each segment $j$ |
| A | The area enclosed by the power requirement at any time p(t) and pmax |
| AMS | Advanced metering system |
| B | The accumulated gain over a month for being flexible given a maximum power tariff; Equals G × Ke |
| BESS | Battery energy storage systems |
| BMS | Battery management system |
| c | Curtailment coefficient, defining the relation between the period for curtailment and the total period (in relation to Figure 10) |
| cp | Acuteness coefficient |
| Ec | Energy requirement including compensation for loss |
| Ec′ | Energy requirement without loss |
| *Emax* | Energy required to sustain the duration of an intervention period |
| EV | Electric vehicle |
| G | Gain: the savings or monetary benefit achieved by being flexible or discharging the battery |
| G′ | Required minimum compensation for taking part in the flexibility scheme, based on V2G/B |
| *I* | Investment: the total cost of the energy flexibility instrument, such as a battery |
| *i* | Interest rate |
| *J* | Number of equally sized segments having an energy capacity of $e_j$ |
| *k* | Number of time periods |
| Ke | Unit cost per kW per month for the largest peak over a given period $T$ (€/kWmonth) |
| l | Battery lifetime |
| m | Coefficient referring to the battery (or material) properties and their resilience to fatigue |
| N | Maximum payback time on an investment |
| *n* | Repayment period (in months) |
| NPV | Net present value |
| *P* | Unit price for energy storage (€/kWh) |
| p(t) | Power demand at any time step t with no intervention |
| *P′* | Initial "unit goodwill capital" of an EV owner |

| pmax | Power ceiling that determines the scale of intervention |
| $\rho(n)$ | Decay function which decreases the goodwill over time |
| R-factor | Equals y/A |
| SOC | State-of-charge |
| T | required peak shaving duration |
| ToU | Time-of-use |
| V2G/B | Transfer of energy from vehicle to grid or from vehicle to building |
| $y$ | Required peak curtailment that defines a power ceiling $y(t) = p(t) - pmax$ |

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
