# Peer review of "E-Mobility and Batteries—A Business Case for Flexibility in the Arctic Region"

_wevj, doi:10.3390/wevj14030061_

Round 1
Reviewer 1 Report
This paper investigates a use case for battery energy storage to regulate peak performance in a small grid system. Therefore, the authors chose a setting in the Nordic region. The paper introduces a metric, namely the "R-factor", which should show a relationship between the energy density and power density of the required battery system. The title explicitly mentions e-mobility, which is only given by naming a use case of snowmobiles in the paper. There is no case shown explicitly referring to vehicles or vehicle-to-X applications. The paper more generally indicates the overall benefit of peak performance regulation and how beneficial it could be regarding cost optimisation. While focusing on costs, battery degradation is not included in the study, which significantly changes the system behaviour over the lifetime and should be investigated here.
The concept of this work is generally interesting, but the structure and given information need to show useful or new information that benefits research, which is not delivered here. Overall, it is hard to follow the central topic, and the different given information should need to be more connected.
The paper cannot be accepted in its current form.
Further Comments:
- Variables in equations or graphics are not introduced.
- Abbreviations are not introduced or introduced but not used properly. (e.g. l.113 second time “V2B” introduced)
- The abstract should mention the main results to give the reader a quick overview of the paper.
- Very poor graphic design (e.g. Fig.11 even shows a curl in the data series)
o Appears to be drawn by hand
- When introducing a metric, give accurate information about the used parameters. (equation R = y/A)
- Inconsistency in the used units. (e.g. currency is referred to as Euro or NOK)
- The energy and power density are not the correct words here, but the overall energy of the battery and the maximum power contribution
- Sources are poorly used. (e.g. for studies or values used)
- The overall requirements/scope of the system are not introduced.
Author Response
Thank you for your good and veryrelevant comments. We have improved the paper accordingly.

Reviewer 2 Report
Comments to the Author:
The manuscript is interesting to me and well organized but some improvements should be made and some technical challenges addressed.
1. It is recommended that abbreviations be provided at the beginning of the paper to define all terms clearly.
2. It is recommended to provide a nomenclature at the beginning of the paper to define all variables clearly.
3. In lines 68&89, There is a problem with the number format in this article, please check it carefully.
4. Also in lines 178&277, There is a problem with the number format in this article, please check it carefully.
5. In lines 89,147,157 and others, “2” should be written as the lower index as CO2 → CO2.
6. In line 258, what mean the abbreviation NOK, please add its meaning.
Finally, with plausible technical explanations regarding these gaps found, this paper could be published.
Author Response
Thank you for your good and very relevant comments. We have improved the paper accordingly.

Reviewer 3 Report
This paper presents an analysis on the feasibility and flexibility of the integration of e-mobility, including batteries, in the arctic region.
The proposed approach, the content of the article and the results presented here are interesting, considering the future of e-mobility and batteries integration.
The authors should take into consideration the following issues:
- Page 2: a state of the art on the subject approached in the paper is missing. The authors should and a special section dedicated to this aspect.
- Page 2 and 3: define quantities R, y and A for the graphs in Figures 1 and 2.
- Page 7: define B in eq. (11), and explain how one can obtain G=y * K_e from this equation!
Author Response

(The authors gave the same response as above.)

Round 2
Reviewer 1 Report
- The abstract should mention the relevant results to give the reader a quick overview of the paper, not only general phrases but tangible values.
- Still poor graphic design (e.g. Fig.11 even shows a curl in the data series)
o Often appears to be drawn by hand
o Check alignments
- The overall requirements/scope of the system are not introduced.
- Fig 6. Please readjust the plots for a better understanding
o Why choose particular savings per month, and how do these get in line with the max capacity
- Example given in Fig. 11 how does this align with the topic?
- Keyword “Snowmobile”
o never necessary in the paper
o talking about battery capacities of more than 20kWh, how does this align?
- Considering the business case – how does this align with charging/usage of the battery (especially for mobile applications)?
- Chapter 3 “Results”
o Add the relevant assumptions which lead to the resulting capacities – in 3.1
o Better introduce the scope of the work – covering V2G application but only speaking of price per kWh of the battery (here, the specific vehicle needs to be considered as well?)
Further Comments:
- l. 331, l.344, l.547 an abbreviation is introduced separately in the text
- p. 9 why mention eq.11 if not used afterwards (eq 12/13)
- l. 388/ l.268 sources for the price tags
Author Response
Thank you for the in-depth comments. We have attemted to reply promptly and clearly.

Reviewer 3 Report
No more suggestions.
Author Response
Thank you for the feedback. We have improved the article also based on another reviewer's comment.
Round 3
Reviewer 1 Report
- Results in 3.1 still need to be explained. What equation is used to obtain, e.g. 52.1kWh in l.393
o Probably, set a significant example
- After changes, readjust the layout (e.g. l361 the graphic and on the next page l.362 the title)
- The "wrench and cut" follows the principle of the demand side management, right?
- It needs to be given how long the V2G application is available per day. Can the battery be used all day? Also, a short description of the recharge and the costs going along with it needs to be included.
o Clarify how much energy per charge/discharge cycle can be used for the V2G/B application
o If the scope relies more on stationary applications, turn that out better
Minor Comments:
l.187 kWh/h = kW
eq. 11 und eq. 12 use same font style (e.g. G)
Author Response
We are greatful for the comments made and have addressed all of them in the revised version of the manuscript.

Round 4
Reviewer 1 Report
Still the switch between the business case (stationary battery storage) and V2G is not clear. The business case, especially the wrench and cut relies on the smallest possible battery here. For V2G, as mentioned in the introduction, the energy needed for the external application is not as relevant, therefore the business case should be different here.
- E.g. l 398 “Hence, it is necessary to discount the battery investment accordingly.” Again, for V2G it is not a particular battery investment but an investment on the mobility.
Minor comments:
· Check referencing of facts, e.g. Table 2 or
· l.587 5kW discharge capacity (Ah – capacity; W – power)
· l.587ff. the new added lines – how does the given values get in line with each other/Table 2
· Table 2 alignment last column (Tables overall: check for “.” and “,” as separator)
· Abbreviations, e.g. l.412
· Some mistakes in language/spelling, e.g. l.413 “With a potential monthly savings”
· Spacing between sentences (appears to be more than one spacing multiple times)
Author Response
The autors thank the reviewer for the good comments. They have now been adressed in the revised version of the manuscript.
